# Diabetic Ketoacidosis Associated with Thyroxine (T_4_) Toxicosis and Thyrotoxic Cardiomyopathy

**DOI:** 10.3390/medicina54060093

**Published:** 2018-11-26

**Authors:** Edinson Dante Meregildo Rodriguez, Luis Iván Gordillo Velásquez, José Gustavo Alvarado Moreno

**Affiliations:** 1Department of Emergency & Critical Care, Hospital Regional Lambayeque, Chiclayo, Lambayeque 14012, Peru; 2Department of Internal Medicine, Hospital Nacional Cayetano Heredia, Lima 15102, Peru; luis.gordillo.v@upch.pe; 3Department of Internal Medicine, Hospital Regional Lambayeque, Chiclayo, Lambayeque 14012, Peru; tavoalvarado89@gmail.com

**Keywords:** thyroid storm, thyrotoxicosis, diabetic ketoacidosis, cardiomyopathy

## Abstract

Thyrotoxicosis and diabetic ketoacidosis (DKA) both may present as endocrine emergencies and may have devastating consequences if not diagnosed and managed promptly and effectively. The combination of diabetes mellitus (DM) with thyrotoxicosis is well known, and one condition usually precedes the other. Furthermore, thyrotoxicosis is complicated by some degree of cardiomyopathy in at least 5% de patients; but the coexistence of DKA, thyroxin (T_4_) toxicosis, and acute cardiomyopathy is extremely rare. We describe a case of a man, previously diagnosed with DM but with no past history of thyroid disease, who presented with shock and severe DKA that did not improve despite optimal therapy. The patient evolved with acute pulmonary edema, elevated troponin levels, severe left ventricular systolic dysfunction, and clinical and laboratory evidence of thyroxin (T_4_) toxicosis and thyrotoxic cardiomyopathy. Subsequently, the patient evolved favorably with general support and appropriate therapy for DKA and thyrotoxicosis (hydrocortisone, methimazole, Lugol’s solution) and was discharged a few days later.

## 1. Introduction

Simultaneous presentation of thyrotoxicosis and diabetic ketoacidosis (DKA) have been reported several times, but it is a clinically unusual situation and remains a diagnostic challenge in clinical practice, because both entities share similar manifestations [1,2,3,4,5,6]. Concurrent presentation of DKA, thyroxin (T_4_) toxicosis and thyrotoxic cardiomyopathy is even rarer. Herein, we describe a case of a man with such an extremely unusual presentation.

## 2. Case Presentation

A 42-year-old Hispanic man with diabetes mellitus (DM) type 2 diagnosed five years ago and regularly treated with glybenclamide 5 mg once daily. Otherwise, the patient’s personal and familiar past medical history was unremarkable. He also denied consumption of alcohol, cigarettes, and illegal drugs. He presented to emergency department (ED) on 1 September 2017 with a history of malaise, headache, fever, and generalized body pain during the last 6 days. Three days before admission, he developed watery diarrhea (3 to 4 times a day), tachypnea, and confusion. During the following days, malaise and bowel movement frequency increased (up to 6 times a day). On the day of admission, the patient became drowsy, dyspneic, and looked very ill. 

Physical examination: Body weight 50 kg, Body Mass Index 16.8 kg/m^2^, blood pressure: 60/30 mmHg, respiratory rate: 32 bpm, heart rate: 78 bpm, axillar temperature: 36 °C, SatO_2_: 99% (FiO_2_: 0.21); the patient was severely dehydrated, thyroid gland was not palpable; respiratory system: tachypnea, Kussmaul’s breathing; cardiovascular system: Sweaty mottled skin, and cold extremities with prolonged capillary refill time; neurologic system: Tremor in both hands, patellar and ankle hyperreflexia.

Lab: Hemoglobin 12.9 g/dL, hematocrit 40, platelets 198,000/mm^3^, white blood cells 10,100/mm^3^, segmented neutrophils 70%, bands 1%; serum glucose 460 mg/dL, urea 115 mg/dL, creatinine 1.3 mg/dL. Arterial blood gas analyses (ABG) are shown in Table 1. Liver function tests were normal, except for hypoalbuminemia and hypoproteinemia (3.0 g/dL and 5.18 g/dL, respectively). Urinalysis: pH 5, urine density 1030, leukocytes 2–4/field, granular casts 3+, glucose 3+, ketones 2+, and leucocyte esterase was negative.

Treatment with intravenous normal saline, potassium chloride, insulin infusion, sodium bicarbonate, norepinephrine, and empiric antibiotics (ceftriaxone + metronidazole) was administered. After 10 h of intensive treatment, ABG did not change significantly. So, differential diagnosis was extended and complementary exams were ordered: plasmatic amylase 96 U/L (reference range: 28–100 U/L), lipase 17 U/L (13–60 U/L), TSH 0.024 μIU/L (adult reference range: 0.27–4.2 μIU/L), free-T4 2.16 ng/dL (reference range: 0.82–1.63 ng/dL), total-T3 0.18 ng/mL (reference range: 0.5–2.0 ng/mL), free-T3 0.42 pg/mL (reference range: 2.1–3.8 pg/mL); CPK-MB 101.3 U/L (reference range: 0–25 U/L), total CPK 505.7 U/L (normal: 39–308 U/L); troponin T 25.0 ng/mL (reference range: 0.12–0.6 ng/mL). Based on these results, hydrocortisone 100 mg every 8 h, methimazole 20 mg every 8 h, and Lugol’s solution 10 drops every 8 h, were added on the second day of treatment. Because of shock, beta-blocker was not administered. On the third day of treatment a good evolution was observed, norepinephrine and insulin infusion were discontinued, and NPH insulin was initiated. Based on physical examination, chest x-ray (CXR) (Figure 1 and Figure 2), and progressive decrease in partial oxygen pressure (Table 1) compatible with acute lung edema, intravenous furosemide 20 mg every 12 h was administered for 2 days.

Although ECG was normal, troponin determinations were repeated over the next days and persisted elevated. Echocardiography (5 September 2017) showed borderline pulmonary artery systolic pressure (35 mm Hg), severe LV systolic dysfunction (LV ejection fraction 35%), diastolic dysfunction of restrictive type and global hypokinesia (Figure 3). 

On the 6th day of treatment, hydrocortisone and Lugol’s solution were stopped, and methimazole was reduced by half. The patient was discharged on 7 September 2017, with almost complete recovery. 

We also performed other complementary exams: Twenty-four hour-urine sodium, chloride, and potassium levels were also normal. Direct stool examination showed 0–1 leucocytes/HPF, no parasites, no blood, no erythrocytes. Urine culture and HIV serology was negative. Anti-TPO and anti-thyroglobulin antibodies were negative. Cardiac catheterization and endomyocardial biopsy were not performed. Follow-up echocardiography at 6 months was completely normal. After the patient was discharged, no additional (blood or imaging) study was performed, other than echocardiography.

### Consent for Publication

Written informed consent was obtained from the patients for publication of this article and accompanying images.

## 3. Discussion

DKA is a complex disordered metabolic state characterized by hyperglycemia, ketoacidosis, and ketonuria. DKA usually occurs as a consequence of absolute or relative insulin deficiency that is accompanied by an increase in counterregulatory hormones (glucagon, cortisol, growth hormone, and epinephrine). This hormonal imbalance enhances hepatic gluconeogenesis, glycogenolysis, and lipolysis [1,2,3,4]. 

The most common triggering factors of DKA are infection, insulin therapy omission, and the onset of other diseases. DKA can be triggered by thyrotoxicosis, but the opposite is also true. Hyperthyroidism worsens glycemic control and precipitates DKA in diabetic patients through several mechanisms, such as an increase of intestinal glucose absorption, a decrease in insulin secretion, and a decrease in the peripheral use of glucose due to insulin resistance. Furthermore, thyroid hormones produce an increase in the hepatocyte plasma membrane concentrations of GLUT2, which is the main glucose transporter in the liver, and consequently, contribute to increased hepatic glucose output and abnormal glucose metabolism. Additionally, increased lipolysis is observed in hyperthyroidism, resulting in an increase in free fatty acids (FFA) that stimulates hepatic gluconeogenesis. The increased release of FFA can be partially explained by an enhanced catecholamine-stimulated lipolysis induced by the excess thyroid hormones. Moreover, the nonoxidative glucose disposal in hyperthyroidism is enhanced, resulting in an overproduction of lactate that enters the Cori cycle and promotes further hepatic gluconeogenesis [3,4,5,6].

It is important to note that other rare etiologies must be considered before classifying DKA as idiopathic. Our case emphasizes the importance of a thorough search for precipitating factors in cases of DKA. We should not disregard the search for precipitating factors, especially for those patients who does not have an obvious cause of DKA or a good response to standard DKA therapy [3,4,5]. Although simultaneous presentation of thyrotoxicosis with DKA is infrequent, the implications of this are very important. Both of these entities share several similar features. Thus, it can be easy to overlook this dual diagnosis. Thereby, for patients having DKA, one should seek a thyrotoxicosis, and vice versa. This applies for patients with or without an obvious triggering factor, particularly in women. Another clue in DKA patients that should lead physicians to suspect the possibility of thyrotoxicosis are: (1) the presence of goiter, and (2) persistent tachycardia in any type, with or without fever, even when other thyrotoxic features are absent. Our patient at admission did not present tachycardia, maybe due to profound high anion gap and non-anion gap metabolic acidosis (delta ratio 0.6 mEq/L and delta gap 12) [3,4,5,7,8,9]. The absence of fever, one of the “classical” signs of thyrotoxicosis, in this patient is remarkable. In the largest series of TS, Akamizu et al. found that tachycardia (heart rate ≥ 120) and fever (≥38.0 °C) were present in 76–84% and 42–56% of 282 patients, respectively [10]. Swee and colleagues, reported that severe tachycardia (heart rate > 140 beats per minute) and fever (≥38.2 °C) were present in 60% and 25% of 28 patients with TS, respectively [11].

The diagnosis of thyrotoxicosis and thyroid storm (TS) is based on clinical findings. Standard thyroid function tests cannot differentiate hyperthyroidism from TS. Although, most patients with TS have significant increase in free triiodothyronine (T3) or free thyroxine (T4), the degree of thyroid hormone excess (elevation of T3 and T4, suppression of TSH) typically is not more profound than that seen in patients with uncomplicated thyrotoxicosis. Consequently, the degree of hyperthyroidism is not a criterion for diagnosing TS [12,13]. According to the scoring criteria of Burch and Wartofsky for TS, our patient would have a score of at least 50, which is highly suggestive of thyroid storm. In addition, our patient fulfilled TS1criteria for TS of *The Japan Thyroid Association Definition and Diagnostic Criteria for Thyroid Storm*. That is: Thyrotoxicosis (elevated levels of free triiodothyronine or free thyroxine) plus at least one Central Nervous System manifestation (confusion and drowsy) and one of the following: Fever, tachycardia, CHF, or GI/hepatic manifestation (diarrhea) [12]. But, because the diagnosis of TS may be questionable, because this patient did not meet all typical signs of TS, we preferred to use the term “thyrotoxicosis” instead of “thyroid storm”. “*Thyroxin (T4) toxicosis*” is called to the pattern of low TSH, high serum free T4, and normal T3 concentrations—presented by our patient. It may be found in patients with hyperthyroidism who have a concurrent non-thyroidal illness that decreases extra-thyroidal conversion of T4 to T3. These patients remain hyperthyroid and their serum TSH concentrations are low; with recovery from the non-thyroidal illness, serum T3 concentrations rise unless the hyperthyroidism is recognized and treated [14].

In most cases of thyrotoxicosis and TS, a precipitating event can be identified. DKA is one of the known precipitants of thyrotoxicosis and TS. Some studies have reported that 20 to 35% of patients with thyrotoxicosis or TS can have no identifiable precipitant factor [10,11,12,13]. To the clinician, the development of hyperthyroidism in a patient with preexisting DM is a situation that would generally be expected to lead to difficulties in controlling the DM and potentially to DKA may develop [7,8,9,15].

With respect to the cardiomyopathy observed in this patient, the most probable etiology is thyrotoxic cardiomyopathy, given the hormonal profile and rapid improvement with anti-thyroid therapy. Other causes of acute reversible cardiomyopathy such as sepsis-induced cardiomyopathy, Takotsubo cardiomyopathy, and other forms of non-Takotsubo stress-induced cardiomyopathy are quite unlikely because we did not see any typical echocardiographic patterns [16,17,18,19]. According to the World Health Organization, thyrotoxic cardiomyopathy defines a myocardial damage caused by the toxic effects of abundant thyroid hormones. But until now, there has been no established diagnostic criteria of the term “thyrotoxic cardiomyopathy” in clinical and research practice. Thyrotoxicosis effects peripheral circulation (increased circulatory volume, reduced pre-load, pulmonary hypertension) and has direct cardiac effects including arrhythmias, alteration of myocardial contractility, left ventricular hypertrophy and cardiomyopathy. The mechanisms are poorly understood. Conventional treatment for hyperthyroidism usually reverses these cardiac complications, such as in the case of our patient [20,21].

We also do not have any reason to suspect myocarditis in this patient. Diabetic ketoacidosis is typically not associated with myocarditis. However, we did not perform cardiac catheterization nor endomyocardial biopsy to rule out these entities [16,17,18,19].

In this case, it may be impossible to decipher whether there is thyrotoxic crisis triggered DKA or DKA is the precipitating event leading to thyrotoxicosis. Some of the clinical findings of the two clinical entities may overlap and confuse physicians. However, it is important to promptly recognize and aggressively treat the concurrent events for improving chances for a successful outcome. Although the coexistence of these emergency diseases is rare, it should be kept in mind.

## 4. Conclusions

Simultaneous presentation of DKA, thyroxin toxicosis and thyrotoxic cardiomyopathy is extremely rare. In addition, DKA and thyrotoxicosis share several similar features and thus, they can be easy to overlook. Thereby, as a simple rule of thumb, for patients having DKA, one should seek a thyrotoxicosis, and vice versa.

## Figures and Tables

**Figure 1 medicina-54-00093-f001:**
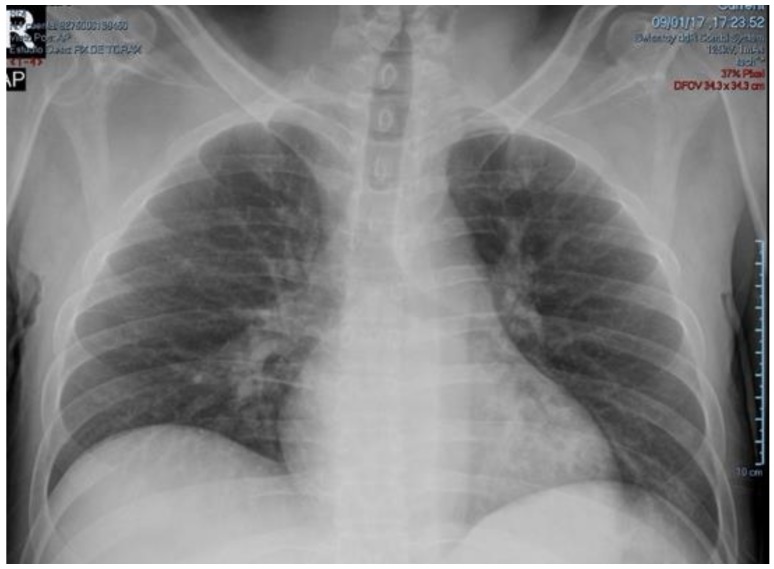
CxR at admission showing no signs of alveolar infiltrates or pleural effusions.

**Figure 2 medicina-54-00093-f002:**
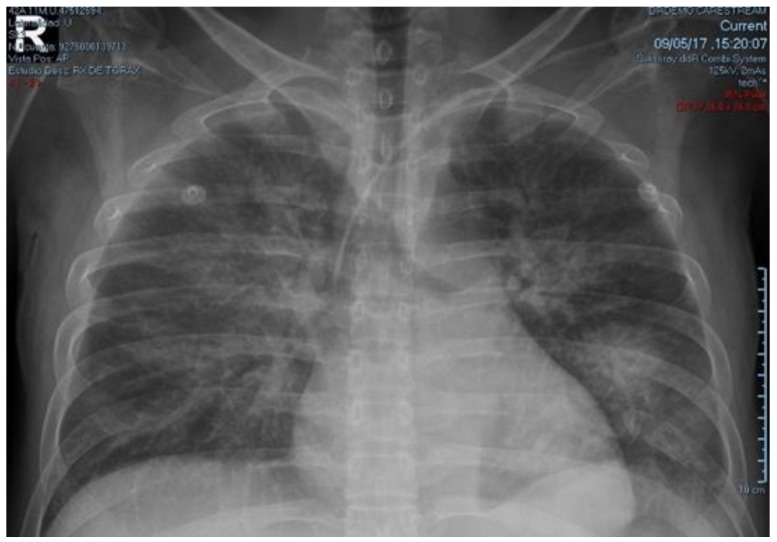
CxR taken four days after admission showing bilateral alveolar infiltrates suggestive of acute pulmonary edema.

**Figure 3 medicina-54-00093-f003:**
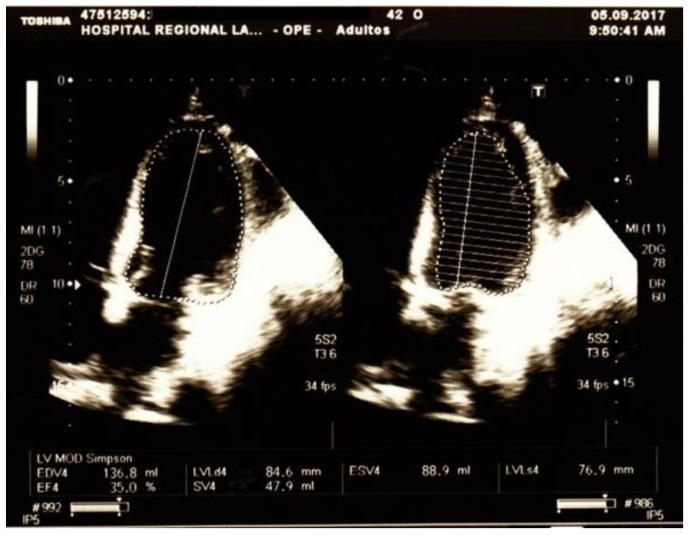
Echocardiography taken on the fifth day after admission showing borderline pulmonary artery systolic pressure (35 mm Hg), severe LV systolic dysfunction (LV ejection fraction 35%), diastolic dysfunction of restrictive type and global hypokinesia.

**Table 1 medicina-54-00093-t001:** Arterial Blood Analysis.

	Reference Values (RV)	1 September 2017; 14:15	2 September 2017; 00:15	3 September 2017; 03:15	3 September 2017; 22:30
pH	7.35–7.45	6.99	6.99	7.42	7.49
pO_2_	75–100 mmHg	160	116	52	55
PCO_2_	35–45 mmHg	12.5	16.2	21.8	22.0
HCO_3_	21–28 mmol/L	3.0	3.0	11.0	19.0
Na^+^	135–145 mmol/L	133	142	140	141
K^+^	3.5–5 mmol/L	3	3.1	3.3	2.6
Cl^−^	102–109 mmol/L	106	120	119	114
Ca^2+^	2.2–2.6 mmol/L	1.3	1.28	1.29	1.21
GAP	10 ± 2 mmol/L	24	19	12.1	8.0
Base deficit	−2–+2	−28.3	−27.6	−10.0	−6.0

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
