# Peer review of "Diabetic Ketoacidosis Associated with Thyroxine (T4) Toxicosis and Thyrotoxic Cardiomyopathy"

_medicina, 2018, doi:10.3390/medicina54060093_

Reviewer 1 Report

Major concerns. 

1.       The novelty and uniqueness of data presented.

The authors propose that the uniqueness and novelty of the research were in the identification and description of the clinical case of the coexistence of DKA, thyroid storm, and cardiomyopathy, which had never been reported before.

However, according to the literature, hyperthyroidism worsens glycemic control in diabetic patients and may precipitate diabetic ketoacidosis, while women with diabetes have a higher prevalence of Graves' disease. [Eva Sola, Carlos Morillas, Stefania Garzon, M Gómez-Balaguer.  Acta Diabetol. 2002 Dec;39(4):235-7.]. With that, even in primary hyperthyroidism in 6% of patients the initial clinical manifestation of the disease is congestive heart failure (CHF), and at least one third of these patients develop persistent dilated cardiomyopathy [Siu CW, Yeung CY, Lau CP, Kung AWC, Tse HF. Heart; 2007: 93: 483–7.]. Similar results were described by Dahl, P et al. (2008)—about 6% of thyrotoxic individuals develop symptoms of heart failure, and less than 1% develop dilated -cardiomyopathy with impaired left ventricular systolic function [Dahl P, Danzi S, Klein I. Thyrotoxic cardiac disease. Curr Heart Fail Rep. 2008: 5170-6.]. Moreover, the cases of Takotsubo cardiomyopathy [Sarapultsev, P. A., & Sarapultsev, A. P. (2016). International journal of cardiology, 221, 698-718.] in diabetic ketoacidosis were described [Nanda S, Longo S, Bhatt SP, Pamula J, Sharma SG, Dale TH. Ann Clin Biochem. 2009 May;46(Pt 3):257-60.],  and Takotsubo was recommended to be considered for any patient presenting with heart failure after treatment of DKA [Jed H. Meyers and Irl B. Hirsch (2017) AACE Clinical Case Reports: Winter 2017, Vol. 3, No. 1, pp. e44-e45.]. As a result, the foregoing does not allow the authors to speak about the uniqueness of that, although quite a rare phenomenon.

2.       The presence of thyroid storm

While, both ketoacidosis and thyrotoxic crisis might independently cause cardiomyopathy, the evidence of a simultaneous effect of those conditions on myocardium would be of great value.

With that, there is an ample evidence of the presence of ketoacidosis in patient: a gradual onset of the disease, a clinical picture with the development of watery diarrhea (3-4 times a day), tachypnea and confusion, severe dehydration, blood pressure: 60/30 mm Hg, respiratory rate : 32 per minute, heart rate: 78 beats / min, axillary temperature: 36 ° C and cold extremities with a long capillary filling time, with the  serum glucose 460 mg / dL were described. The presence of the heart damage was also described: the rise of CPK-MB up to  101.3 U / l (normal range: 0-25 U / l), total CPK—up to 505.7 U / l (normal: 39-308 U / l); troponin T—up to 25.0 ng / ml (normal range: 0.12-0.6 ng / ml) and by EchoCG data: the signs of severe LV systolic dysfunction (LV ejection fraction 35%), the resistive-type diastolic dysfunction and the global hypokinesia.

However, there are no data about the possible reasons for the development of a thyrotoxic crisis, which episodes mostly occur either in patients with known hyperthyroidism whose treatment has been stopped or become ineffective, or in patients with untreated mild hyperthyroidism who have developed an intercurrent illness (such as an infection) [Klubo-Gwiezdzinska, Joanna; Wartofsky, Leonard (March 2012). "Thyroid emergencies". Medical Clinics of North America. 96 (2): 385–403.]. Also, based on the data presented, the patient described had no typical signs of a thyrotoxic crisis, such as acute onset of symptoms of hyperthyroidism (fast heart rate, anxiety, agitation), accompanied by other features, such as fever (temperature above 40 ° C / 104 ° F ), hypertension [Gardner, DG (2017)]; there is no significant increase in free triiodothyronine (T3) or free thyroxine (T4), typical for the thyroid storm [Akamizu, Takashi; Satoh, Tetsurou; Isozaki, Osamu; Suzuki, Atsushi; Wakino, Shu; Iburi, Tadao; Tsuboi, Kumiko; Monden, Tsuyoshi; Kouki, Tsuyoshi.Thyroid. 2012 Jul; 22 (7): 661-79.].

With that, the diagnosis of thyroid storm (lines 121-123) remains controversial enough and requires additional information to be confirmed.

Conclusion: The introduction and the discussion part of the manuscript should be revised. English editing is suggested. Possibly, the title of the manuscript should be changed. 

Minor concerns.

1.       The description of the patient as “young men” (line 29), while it age is 42 (line 32) is controversial.

2.       Minor language editing is needed.

Author Response

Response to Reviewer 1 Comments

Major concerns.

Point 1.       The novelty and uniqueness of data presented.

The authors propose that the uniqueness and novelty of the research were in the identification and description of the clinical case of the coexistence of DKA, thyroid storm, and cardiomyopathy, which had never been reported before.

However, according to the literature, hyperthyroidism worsens glycemic control in diabetic patients and may precipitate diabetic ketoacidosis, while women with diabetes have a higher prevalence of Graves' disease. [Eva Sola, Carlos Morillas, Stefania Garzon, M Gómez-Balaguer.  Acta Diabetol. 2002 Dec;39(4):235-7.]. With that, even in primary hyperthyroidism in 6% of patients the initial clinical manifestation of the disease is congestive heart failure (CHF), and at least one third of these patients develop persistent dilated cardiomyopathy [Siu CW, Yeung CY, Lau CP, Kung AWC, Tse HF. Heart; 2007: 93: 483–7.]. Similar results were described by Dahl, P et al. (2008)—about 6% of thyrotoxic individuals develop symptoms of heart failure, and less than 1% develop dilated -cardiomyopathy with impaired left ventricular systolic function [Dahl P, Danzi S, Klein I. Thyrotoxic cardiac disease. Curr Heart Fail Rep. 2008: 5170-6.]. Moreover, the cases of Takotsubo cardiomyopathy [Sarapultsev, P. A., & Sarapultsev, A. P. (2016). International journal of cardiology, 221, 698-718.] in diabetic ketoacidosis were described [Nanda S, Longo S, Bhatt SP, Pamula J, Sharma SG, Dale TH. Ann Clin Biochem. 2009 May;46(Pt 3):257-60.], and Takotsubo was recommended to be considered for any patient presenting with heart failure after treatment of DKA [Jed H. Meyers and Irl B. Hirsch (2017) AACE Clinical Case Reports: Winter 2017, Vol. 3, No. 1, pp. e44-e45.]. As a result, the foregoing does not allow the authors to speak about the uniqueness of that, although quite a rare phenomenon.

Response 1: According to information given in Response 2 (see below), we have preferred to change the term “thyroid storm” used in the original paper by “thyrotoxicosis” or “thyroxin (T4) toxicosis”. As a matter of fact, the coexistence of DKA, thyroxin (T4) toxicosis, and acute cardiomyopathy is extremely rare, and to the best of our knowledge it has never previously been reported. Even though, we have preferred to omit this statement in the final manuscript.

Point 2.       The presence of thyroid storm

While, both ketoacidosis and thyrotoxic crisis might independently cause cardiomyopathy, the evidence of a simultaneous effect of those conditions on myocardium would be of great value.

With that, there is an ample evidence of the presence of ketoacidosis in patient: a gradual onset of the disease, a clinical picture with the development of watery diarrhea (3-4 times a day), tachypnea and confusion, severe dehydration, blood pressure: 60/30 mm Hg, respiratory rate : 32 per minute, heart rate: 78 beats / min, axillary temperature: 36 ° C and cold extremities with a long capillary filling time, with the  serum glucose 460 mg / dL were described. The presence of the heart damage was also described: the rise of CPK-MB up to  101.3 U / l (normal range: 0-25 U / l), total CPK—up to 505.7 U / l (normal: 39-308 U / l); troponin T—up to 25.0 ng / ml (normal range: 0.12-0.6 ng / ml) and by EchoCG data: the signs of severe LV systolic dysfunction (LV ejection fraction 35%), the resistive-type diastolic dysfunction and the global hypokinesia.

However, there are no data about the possible reasons for the development of a thyrotoxic crisis, which episodes mostly occur either in patients with known hyperthyroidism whose treatment has been stopped or become ineffective, or in patients with untreated mild hyperthyroidism who have developed an intercurrent illness (such as an infection) [Klubo-Gwiezdzinska, Joanna; Wartofsky, Leonard (March 2012). "Thyroid emergencies". Medical Clinics of North America. 96 (2): 385–403.]. Also, based on the data presented, the patient described had no typical signs of a thyrotoxic crisis, such as acute onset of symptoms of hyperthyroidism (fast heart rate, anxiety, agitation), accompanied by other features, such as fever (temperature above 40 ° C / 104 ° F ), hypertension [Gardner, DG (2017)]; there is no significant increase in free triiodothyronine (T3) or free thyroxine (T4), typical for the thyroid storm [Akamizu, Takashi; Satoh, Tetsurou; Isozaki, Osamu; Suzuki, Atsushi; Wakino, Shu; Iburi, Tadao; Tsuboi, Kumiko; Monden, Tsuyoshi; Kouki, Tsuyoshi.Thyroid. 2012 Jul; 22 (7): 661-79.].

With that, the diagnosis of thyroid storm (lines 121-123) remains controversial enough and requires additional information to be confirmed.

Conclusion: The introduction and the discussion part of the manuscript should be revised. English editing is suggested. Possibly, the title of the manuscript should be changed.

Response 1:

First of all, we must say that, in order to be concise enough and not to extend excessively this paper, in the original version, we have omitted important information with respect to the Physical Exam, that in this new version was added (see pages 44 to 49).

We agree with the reviewer point of view that the diagnosis of thyroid storm (TS) could be controversial, and as we stated in Introduction (line 32) and Discussion (lines 118 and 119), both of these entities (TS and DKA) share several similar features and it can be easy to overlook this dual diagnosis. So, we could no certainly affirm that this is a genuine case of TS. But strictly, our patient fulfilled TS1criteria for TS of The Japan Thyroid Association Definition and Diagnostic Criteria for Thyroid Storm. That is: thyrotoxicosis (elevated levels of free triiodothyronine or free thyroxine) plus at least one CNS manifestation (confusion and drowsy) and one of the following: fever, tachycardia, CHF, or GI/hepatic manifestation (diarrhea) [1].

In addition, as we explained in Discussion (Lines 150 and 151) in this case, it may be impossible to ascertain whether there was thyrotoxic crisis triggered DKA or DKA was the precipitating event leading to thyroid storm. In addition, other studies have reported that 20 to 35% of patients with TS can have no identifiable precipitant factor [1-3].

In Discussion (lines 123 to 125) we have explained one possible reason why our patient did not present typical signs of TS, such as acute tachycardia (profound metabolic acidosis). But, the absence of fever in this patient is remarkable. In the largest series of TS, Akamizu et al. found that tachycardia (heart rate ≥120) and fever (38.0°C) were present in 76-84% and 42-56%, respectively, of 282 patients [2]. Swee and colleagues, reported that severe tachycardia (heart rate >140 beats per minunte) and fever (38.2°C) were present in 60% and 25%, respectively, of 28 patients with TS [4].

With respect to T4 and T3, although, most patients with TS have significant increase in free triiodothyronine (T3) or free thyroxine (T4), the degree of thyroid hormone excess (elevation of T3 and T4, suppression of TSH) typically is not more profound than that seen in patients with uncomplicated thyrotoxicosis. So, standard thyroid function tests cannot differentiate hyperthyroidism from thyroid storm, as we stated in Discussion (Lines 126 and 127). Consequently, the degree of hyperthyroidism is not a criterion for diagnosing TS [1-3].

The pattern of low TSH, high serum free T4, and normal T3 concentrations —presented by our patient— is called T4-toxicosis. It may be found in patients with hyperthyroidism who have a concurrent nonthyroidal illness that decreases extrathyroidal conversion of T4 to T3. These patients remain hyperthyroid and their serum TSH concentrations are low; with recovery from the nonthyroidal illness, serum T3 concentrations rise unless the hyperthyroidism is recognized and treated [5].

In consequence, because the diagnosis of TS may be questionable, we preferred to use the term “thyroxin (T4) toxicosis” or just “thyrotoxicosis” instead of “thyroid storm”.

Minor concerns.

Point 1.       The description of the patient as “young men” (line 29), while it age is 42 (line 32) is controversial.

Response 3: We have eliminated this adjective in line 34.

Point 2.       Minor language editing is needed.

Response 4: We have revised and checked misspelling errors.

ADITIONAL REFERENCES.

1.      Akamizu T. Thyroid Storm: A Japanese Perspective. Thyroid. 2018; 28(1): 32–40. doi:  10.1089/thy.2017.0243.

2.      Akamizu T, Satoh T, Isozaki O, Suzuki A, Wakino S, Iburi T, Tsuboi K, Monden T, Kouki T, Otani H, Teramukai S, Uehara R, Nakamura Y, Nagai M, Mori M; Japan Thyroid Association. Diagnostic criteria, clinical features, and incidence of thyroid storm based on nationwide surveys. Thyroid. 2012; 22(7):661-79. doi: 10.1089/thy.2011.0334.

3.      Idrose AM. Acute and emergency care for thyrotoxicosis and thyroid storm. Acute Med Surg. 2015; 2(3):147-157. doi: 10.1002/ams2.104.

4.      Swee du S, Chng CL, Lim A. Clinical characteristics and outcome of thyroid storm: a case series and review of neuropsychiatric derangements in thyrotoxicosis. Endocr Pract. 2015 Feb;21(2):182-9. doi: 10.4158/EP14023.OR.

5.      Caplan RH, Pagliara AS, Wickus G. Thyroxine toxicosis. A common variant of hyperthyroidism. JAMA. 1980;244(17):1934.

6.      Satoh T, Isozaki O, Suzuki A, Wakino S, Iburi T, Tsuboi K, Kanamoto N, Otani H, Furukawa Y, Teramukai S, Akamizu T. 2016 Guidelines for the management of thyroid storm from The Japan Thyroid Association and Japan Endocrine Society (First edition). Endocr J. 2016 Dec 30;63(12):1025-1064. doi: 10.1507/endocrj.EJ16-0336.

Reviewer 2 Report

The manuscript entitled “Diabetic ketoacidosis associated with thyroid storm and thyrotoxic cardiomyopathy” by E.D.M. Rodriguez and colleagues is a case report describing a rare presentation of DKA, thyroid storm and thyrotoxic cardiomyopathy in a patient.

As correctly cited by the authors themselves, similar associations have been described in the past; still, the present case appears as a relevant information worth being disclosed to the audience in the field.

The text has been analyzed for plagiarism, detecting a borderline/acceptable level of similarity with existing texts (19%), below critical threshold (analysis performed by Magister – Compilatio.net).

Overall, the report is well described and discussed; some minor revisions are here proposed:

- Introduction: 

Line 26: the DKA acronym (even if explained in the abstract) should be expanded in the introduction section too.

- Case-presentation: 

Line 32: ethnic information on the patient is missing. Family history is missing. Social information and remote pathological history is missing. Body weight and BMI are not reported.

Line 85: follow-up has been performed with echocardiography; is there any blood analysis available at 6 months (or later)?

- Discussion:

Line 119: does “form” stand for “from”?

Line 126: does “We” stand for “With”?

Author Response

Response to Reviewer 2 Comments

Introduction:

Point 1-Line 26: the DKA acronym (even if explained in the abstract) should be expanded in the introduction section too.

Response 1: The complete description of DKA (diabetic ketoacidosis) is added, now in Line 30.

Case-presentation:

Point 2- Line 32: ethnic information on the patient is missing. Family history is missing. Social information and remote pathological history are missing. Body weight and BMI are not reported.

Response 2: We have provided additional information about the patient: ethnic, family, social and remote pathological history; and body weight and BMI were also reported between line 36 to 39.

Point 3- Line 85: follow-up has been performed with echocardiography; is there any blood analysis available at 6 months (or later)?

Response 3: After the patient was discharged, no additional (blood or imaging) study was performed, other than echocardiography. We have included this additional information about the patient evolution in line 93 to 95.

Discussion:

Point 4- Line 119: does “form” stand for “from”?

Response 4: Spelling mistake was corrected, see Line 133.

Point 5- Line 126: does “We” stand for “With”?

Response 5:  Spelling mistake was corrected, see Line 157.

Round  2

Reviewer 1 Report

Dear authors, thank you for the work done!